# Partners at Care Transitions: exploring healthcare professionals' perspectives of excellence at care transitions for older people

Ruth Baxter,[1] Jane O'Hara,[1,2] Jenni Murray,[1] Laura Sheard,[1] Alison Cracknell,[3] Robbie Foy,[4] John Wright,[1] Rebecca Lawton[1,5]

[1]Yorkshire Quality and Safety Research Group, Bradford Institute for Health Research, Bradford, UK
[2]Leeds Institute of Medical Education, University of Leeds, Leeds, West Yorkshire, UK
[3]Leeds Centre for Older People's Medicine, Leeds Teaching Hospitals NHS Trust, Leeds, UK
[4]Leeds Institute of Health Sciences, University of Leeds, Leeds, UK
[5]School of Psychology, University of Leeds, Leeds, UK

**Correspondence to**
Dr Ruth Baxter;
ruth.baxter@bthft.nhs.uk

## ABSTRACT

**Introduction** Hospital admissions are shorter than they were 10 years ago. Notwithstanding the benefits of this, patients often leave hospital requiring ongoing care. The transition period can therefore be risky, particularly for older people with complex health and social care needs. Previous research has predominantly focused on the errors and harms that occur during transitions of care. In contrast, this study adopts an asset-based approach to learn from factors that facilitate safe outcomes. It seeks to explore how staff within high-performing ('positively deviant') teams successfully support transitions from hospital to home for older people.

**Methods and analysis** Six high-performing general practices and six hospital specialties that demonstrate exceptionally low or reducing 30-day emergency hospital readmission rates will be invited to participate in the study. Healthcare staff from these clinical teams will be recruited to take part in focus groups, individual interviews and/or observations of staff meetings. Data collection will explore the ways in which teams successfully deliver exceptionally safe transitional care and how they overcome the challenges faced in their everyday clinical work. Data will be thematically analysed using a pen portrait approach to identify the manifest (explicit) and latent (abstract) factors that facilitate success.

**Ethics and dissemination** Ethical approval was obtained from the University of Leeds. The study will help develop our understanding of how multidisciplinary staff within different healthcare settings successfully support care transitions for older people. Findings will be disseminated to academic and clinical audiences through peer-reviewed articles, conferences and workshops. Findings will also inform the development of an intervention to improve the safety and experience of older people during transitions from hospital to home.

## BACKGROUND

Healthcare services in the UK are increasingly being delivered within community settings and the average length of a hospital stay is now 2 days shorter than it was 10 years ago.[1] This aligns with patient preferences to be at home and benefits hospitals by improving flow and

### Strengths and limitations of this study

► Instead of focusing on error and harm, this study adopts a 'Safety-II' perspective and applies the positive deviance approach to explore how healthcare professionals support *successful* transitions of care for older patients.
► Perspectives about how safe transitional care is delivered will be gathered from a wide range of multidisciplinary clinical staff working in a variety of different healthcare settings.
► This will allow an exploration of both the commonalities and differences that exist across and between different contexts.
► Problems commonly associated with routinely collected data may limit the extent to which healthcare teams can truly be considered high performers.
► Focus group and interview methods may not uncover the underlying, latent factors that contribute to successful transitions of care as they rely on the ability of staff to identify and verbalise the ways in which they succeed.

enabling them to treat more people. However, shorter lengths of stay mean patients often leave hospital with ongoing care needs that are directly related to their recent hospitalisation, such as wound care treatment, changes to, or monitoring of, new medications and limited mobility associated with deconditioning and inactivity. The risk of complications and the discontinuity of care that occurs between primary and secondary healthcare services increase a patient's vulnerability to things going wrong.[2] As many as one in five patients experience an adverse event during the transition from hospital to home, 62% of which could be prevented or minimised.[3] Emergency hospital readmission rates are the only routinely collected measure of quality at transitions within the English National Health Service (NHS). Since 2012/2013 emergency readmission rates have increased

**BMJ**

by 22.8% from 372 805 to 457 880[4] and, although national statistics do not assess preventability, research studies estimate that around 30% of all readmissions are potentially avoidable.[5–7]

Transitions of care are particularly risky for older people with more complex health and social care needs (eg, comorbidities, frailty and cognitive impairments). Given that older people are more likely to be readmitted to hospital[8–10] and that this population accounts for the majority of NHS patients,[1] improving the quality and safety of care for older people during transitions is a priority for the UK's Department of Health. Furthermore, the financial penalties incurred by NHS trusts (secondary care providers) exert economic pressure to reduce emergency hospital readmissions.[11]

The traditional approach to safety management within healthcare, known as Safety-I, has been to focus on past errors and harm.[12] Most previous research exploring healthcare professionals' perceptions about the quality and safety of care transitions has adopted this Safety-I approach by highlighting the fundamental and universal system flaws. These include communication breakdowns across professional groups and care settings (exacerbated by poor understanding of mutual roles in the transition), a lack of accountability and poor communication with the patient.[13–17] Research has sought to understand what goes wrong at transitions of care in order to provide top down guidance on potential solutions. Although useful, this deficit-based approach only tells us about the *absence* of safety. It says nothing about how safe care is delivered successfully, what the solutions to problems may be or whether a chosen solution is likely to be effective.

### Adopting an asset-based approach to safety management

Over the past 5 years, a more positive approach to safety management, commonly known as Safety-II, has been proposed.[12 18] This approach recognises that healthcare 'goes right' far more often than it 'goes wrong' and that exploring and maximising how *safe* (rather than unsafe) care is delivered may help to reduce errors and improve safety.

Furthermore, healthcare organisations are considered to be complex adaptive systems.[19] Autonomous healthcare professionals interact across multiple networks within dynamic, non-linear and self-organising systems that change over time.[20] Given this complexity, the top down, linear solutions of Safety-I may be less effective. Healthcare professionals interpret and tailor guidelines and procedures—their actions are influenced by networks, unpredictable working environments and the competing bottom up, informal rules to which they work.[21] As a result, healthcare teams and organisations demonstrate resilience—they respond, monitor, learn and anticipate. It is this intrinsic ability to adjust to changes and disturbances that enables them to successfully deliver patient care under expected and unexpected conditions.[22]

In line with the Safety-II perspective, 'positive deviance' provides an asset-based approach to improvement.[23 24]

This approach seeks to identify and learn from individuals, teams or organisations who demonstrate exceptional performance despite facing similar constraints as others. It is increasingly being used within healthcare organisations to improve the quality and safety of patient care,[25 26] for example, in improving hand hygiene behaviours and reducing healthcare-associated infections.[27 28]

Bradley *et al*[24] have proposed a four-stage framework to apply the positive deviance approach within healthcare organisations. Routinely collected data are analysed to identify positive deviants who demonstrate exceptional performance on an outcome of interest (stage 1). Qualitative methods are used to generate hypotheses about how positive deviants succeed (stage 2). These 'positively deviant strategies' are then tested within larger and more representative samples (stage 3) and then disseminated to others with the help of key stakeholders (stage 4).

### Study aim and research questions

This study applies stage 2 of the positive deviance framework.[24] It seeks to explore how high-performing teams (general practices and hospital specialties that demonstrate low or reducing readmission rates over time) successfully deliver safe care to older people and demonstrate resilience during transitions from hospital to home. The following research questions will be addressed:

▶ What strategies do high-performing teams use to deliver safe transitions of care to older people?
▶ What contextual factors are important in facilitating safe transitional care for older people?
▶ What challenges are faced when delivering safe transitional care for older people and how do high-performing teams demonstrate resilience to overcome them?

### Preparatory work to identify high-performing teams

In preparation for this study, stage 1 of the positive deviance framework[24] was conducted to identify high-performing general practices and hospital specialties that demonstrate the lowest or reducing emergency readmission rates over a specified period of time.

The initial study population comprised 151 general practices clustered within five West Yorkshire clinical commissioning groups (CCGs) and 85 hospital specialties clustered within 22 acute NHS Trusts in the North of England. Cardiology, respiratory and medicine for older people specialties were selected as their patient populations are typically older than those of others specialties (eg, general surgery). They were selected following a review of publicly available Hospital Episode Statistics (HES)[29] and a consensus approach with clinical advisors (a geriatrician, a clinical epidemiologist and two general practitioners (GPs)).

For each general practice and hospital specialty, total numbers of hospital discharges and 30-day emergency readmissions for patients aged 75 years and over were extracted from the NHS Secondary Use Services data repository for primary care[30] and HES for secondary

care.[29] In primary care, data were extracted over two time frames: April 2015 to March 2016 (for all five CCGs); and April 2016 to January 2017 (for two CCGs) or to February 2017 (for three CCGs). In secondary care, data were extracted over three time frames: 2013/2014, 2014/2015 and 2015/2016 (November to October for all years). These time frames represent the most recent data that were available in each setting.

The 30-day emergency readmission rates for patients aged 75 years and over were calculated for each general practice and hospital specialty. Statistical Process Control methods were then used to analyse the data in both settings.[31 32] Binomial funnel plots with limits calculated using the Wilson Score method[33] were created at each time point for the five CCGs and three types of hospital specialty. In both settings, control limits were set at 2 and 3 sigma (σ) and high performers were classed as those teams (general practices or hospital specialties) that exceeded these limits. In addition to learning from those who performed the best in the region, bar charts were also plotted using the secondary care data to identify hospital specialties with reducing readmission rates, that is, those with improved performance over time. It was difficult to apply definitive rules to this as data varied markedly between hospital sites and specialties. As such, the researchers discussed the data as a group and selected the high-performing hospital specialties that demonstrated the greatest reduction in readmission rates over the 3-year period. Findings from this preparatory work were used to select a sample of high-performing general practices and hospital specialties for this study.

## METHOD
### Sampling the high-performing teams
A diverse group of high-performing general practices and hospital specialties will be purposively sampled during this study to explore how success is achieved across a range of different healthcare contexts. High-performing general practices that exceed the funnel plot control limits will be purposively sampled using publicly available data for four variables: the practices' total list size (number of patients); a practice level measure of deprivation; the proportion of patients aged 75 years and over and the proportion of patients in a nursing home.[34] General practices will be dichotomised for each variable (above and below average).

In secondary care, the clinical leads from high-performing specialties that either consistently exceeded the funnel plot control limits (over 2 or 3 years), or showed a reduction in readmission rates over time will be approached for a brief telephone conversation (~15 min). The aim of these will be to better understand whether there are any likely explanations, other than their teams performing well, which might explain their high performance (eg, data coding anomalies—see online supplementary file 1 for a topic guide). Specialties will then be purposively sampled using the following criteria:

► To equally represent all three specialties (cardiology, respiratory medicine and geriatric medicine) and the two manifestations of 'high performance' (exceed funnel plot control limits vs improvement over time).
► To represent specialties that care for patients from a range of different socioeconomic backgrounds—as measured by Index of Multiple Deprivation data at NHS Trust level.[35]
► Where relevant, to represent specialties from secondary and tertiary services (eg, cardiology).

High-performing specialties may perform exceptionally well simply because their patients are kept in hospital for longer periods of time and thus have fewer needs on discharge. Length of stay (LoS) data are not publicly available at individual hospital specialty level, therefore, it was not possible to control for this in the preparatory work. To exclude the possibility that high-performing specialties have low readmission rates simply because their patients have exceptionally long hospital stays, LoS data for each participating specialty will be descriptively compared with the publicly available national LoS average for each specialty during the period of 2015/2016.[1]

### Study design and status
A qualitative study will be conducted to explore how multidisciplinary staff within high-performing teams deliver exceptionally safe care to older patients during transitions from hospital to home to prevent unnecessary hospital readmissions. A maximum of six general practices and six hospital specialties that were identified during the preparatory and sampling work described above will be approached to take part in the study. Hospital specialties often contain several individual wards or units, therefore, the researcher will agree with the clinical leads for each specialty which wards to include in the study (based on HES data, patient's age and whether wards are common to most trusts). The preparatory work described above took place between June and August 2017. The study commenced in September 2017 and site recruitment was completed by the end of May 2018. Data collection and analysis will be iterative and flexible according to the needs of the study.

### Participants and recruitment
Multidisciplinary clinical staff who work within the high-performing teams will be recruited to participate in a focus group and/or individual interview. Primary care participants may include GPs, advanced nurse practitioners, practice nurses, healthcare assistants, practice managers, pharmacists, reception and administrative staff. Hospital participants will include doctors, nurses, healthcare assistants, ward clerks, occupational therapists, physiotherapists, pharmacists and discharge coordinators. Most transitions of care involve many different teams and healthcare staff[36] and so this study will also seek to recruit staff from the community teams that support the high-performing general practices and hospital specialties to deliver transitional care to older people (eg, district nurses, community matrons and nurse specialists).

Purposive and opportunity sampling will be used to gather a wide range of perspectives about how safe patient care is delivered. Participants will be identified through the managers in each site and iteratively during data collection. Staff will be approached by either their manager or the researcher. An exact sample size for the study cannot be provided as each high-performing site will vary in size and staff membership. However, as a minimum, at least six staff will be recruited in each site and recruitment will continue until data saturation has been achieved.

## Data collection
### Focus groups and interviews
Most data for this study will be collected using focus groups. At least one focus group lasting up to 60 minutes will be conducted within each high-performing site. Additional focus groups will be held where necessary, for example, in dual-sited general practices or large hospital specialties with multiple wards. As this study seeks to explore care transitions from a number of different perspectives, focus groups will bring together different members of the multidisciplinary team. Professional hierarchies can affect data collection[37] and so, if necessary, focus groups will be conducted within professional groups. In line with guidance,[37] we will seek to recruit around eight participants to each focus group.

If key staff who support transitions of care are unable to attend a focus group (eg, due to shift patterns or locations of work), they will be invited to participate in an individual interview. These will last a maximum of 60 minutes and will be conducted at convenient times and locations. Semi-structured topic guides will be used to facilitate discussions and these will vary by care setting (see online supplementary file 2). Researchers will also make field notes throughout the study to record their observations, for example, of contextual information, team dynamics and culture.

### Observation in hospital specialties
Within secondary care, the researcher will observe one or two staff meetings in each specialty where transitions of care and hospital discharge are discussed (eg, safety huddles, board rounds and multidisciplinary team meetings). Field notes will be used to gather contextual information about how transitions of care are planned. Observations will only involve healthcare professionals; patient-staff interactions and the delivery of direct patient care will not be observed. Specific meetings about transitions of care rarely occur within the community as care delivery is more dispersed. Consequently, observations were not considered feasible in this setting.

General practices and hospital specialties will be offered £200 for their involvement and staff will be given refreshments during focus groups/interviews. Where possible, all focus groups will be facilitated by RB who has a background in health services researcher and is educated to doctorate level. In each setting, RB will be supported by a researcher who has a clinically relevant background—an occupational therapist, a GP registrar and a community nurse. This will facilitate the collection of data from insider and outsider perspectives.[38] All researchers will help to develop the focus group schedules to ensure consistency in their data collection approaches. Researchers will meet regularly to discuss the emerging data.

## Analysis
Focus groups and interviews will be digitally recorded and transcribed verbatim, and field notes will be typed up. A pen portrait approach to analysis will be taken using guidance from previous research.[39] Pen portraits are typically used to synthesise data across different sources and to draw out key data about a particular or 'typical' participant. In this study, pen portraits will be written for each participating team (ie, general practice or ward within a specialty) and will then be analysed thematically following guidance by Braun and Clarke.[40] Pen portraits will be analysed at two levels of abstraction: the manifest (explicit) content such as the strategies and tools that teams use to succeed; and the latent (abstract) content such as team dynamics, culture and ways in which the teams demonstrate resilience. As data will be collected from heterogeneous teams and settings, the analysis will explore the commonalities and differences that exist across and between settings.

To enhance rigour, the researchers will keep a record of the analysis as it develops and will meet regularly with research team members to discuss emergent findings. A proportion of pen portraits will be reviewed by another researcher to ensure that they accurately and consistently reflect the data collected.

## Patient and public involvement
The Yorkshire Quality and Safety Research Group have an active patient panel. This panel contributed to the development of the overarching Partners at Care Transitions (PACT) programme of work, including the focus on older people, the grant proposal and the design of each research study within it (including the one presented here). In addition, a separate patient panel have been recruited specifically for the PACT programme. The PACT patient panel have contributed to the design of this study by sharing their experiences of transitions from hospital to home. Patients and the public will not be involved in the recruitment or conduct of the study as the research focuses on staff perceptions and does not involve patients. However, findings will be discussed with the PACT patient panel so that they can be fully involved in all subsequent aspects of the research programme.

## ETHICS AND DISSEMINATION
### Ethical issues
During the study, all participants will be fully informed about their voluntary involvement. Staff will have a right to withdraw and written informed consent will be gained. Verbal consent will be gained from staff who attend the

observed meetings in secondary care. The confidentiality and anonymity of all participating CCGs, NHS Trusts, hospital specialties, general practices and staff will be maintained. All focus group and interview transcripts will be anonymised and stored securely. Although this study seeks to explore how healthcare professionals deliver safe care to older people during transitions from hospital to home, it is recognised that this is a challenging aspect of a patients' journey where harm can occur.[3] Under our professional duty of care, if there is an immediate threat to the safety of patients or others, researchers will be required to inform a senior member of staff and/or a person responsible for risk management within the organisation.

### Dissemination

Findings from this study will contribute to the ongoing PACT programme of research. A summary of the findings will be given to all participating teams along with the final report if desired (which will be produced for the funding body). A report will also be produced for local CCGs. The study will be written up for publication in peer-reviewed journals and presented at national and international conferences. Workshops and learning events will be conducted to disseminate findings to key stakeholders including representatives from academia, primary, secondary and community healthcare, lay members and third sector organisations.

## DISCUSSION

This study seeks to explore how healthcare professionals, in a variety of different settings, deliver exceptionally safe care to older people during transitions from hospital to home in order to prevent avoidable hospital readmissions. It seeks to understand at both a manifest (explicit) and latent (abstract) level the factors that contribute to a team's success and the ways in which healthcare professionals overcome the challenges that they face while supporting patients during transitions from hospital to home.

The study contributes to the PACT programme of work, which seeks to explore whether greater involvement of patients and their families can improve the safety and experience of older people during transitions from hospital to home and, by doing so, reduce readmissions and NHS costs. The study represents the second of six work packages (WPs) within the programme. The findings will be combined with those from WP1 (in which patient experiences during transitions are explored),[41] the existing literature and WP3 (in which a measure of safety and experience at transitions is developed). These will inform an intervention to involve older patients and their families during transitions from hospital to home to improve safety and experience, which will undergo evaluation in a randomised controlled trial (WP4–WP6).

By better understanding how teams succeed and overcome the challenges of everyday clinical work within existing resources, this study will offer a way of developing intervention strategies that are feasible, sustainable and acceptable to healthcare staff, thereby facilitating spread of the PACT intervention. Moreover, this approach can help explore the contextual features such as intergroup relations, leadership and culture that may be important for success, thus facilitating the smooth implementation of the PACT intervention.

### Strengths and limitations

The study adopts two relatively new approaches to the management and improvement of patient safety—Safety-II[12] and the positive deviance approach.[23 24] In contrast to previous research on errors and harm, this study seeks to understand how clinical teams successfully deliver exceptionally safe care to older people and their families during a transition from hospital to home. However, there are a number of challenges to adopting these approaches. Identification of high-performing general practices and hospital specialties can be challenging. High-performing teams were identified using the most recent data that were available; however, in secondary care these data were still a year old due to publication lags. Although HES data are indirectly standardised by age, sex, method of admission and diagnosis/procedure, there are various biases and confounding factors that could influence a team's performance and limit the ability to make like-for-like comparisons. Coding variations exist between trusts, and commissioning differences across CCGs and NHS Trusts will influence the services and resources that are available to support safe transitions and prevent readmissions (eg, the availability of rehabilitation beds and community nurses to support patients at home). The validity of emergency readmissions data as a quality outcome is also criticised.[42 43] Data include both avoidable and unavoidable readmissions but it is only the former that we seek to reduce. In recognition of these limitations, the study sought to explore high-performing hospital specialities that had reduced their readmission rates over time as well as those that had the lowest readmission rates in the region.

A key strength of this study is the inclusion of multiple perspectives across a variety of healthcare settings to explore how different clinical teams successfully support transitions of care. However, this heterogeneous sample may limit our ability to achieve theoretical saturation. Similarly, focus group and interview methods were chosen to facilitate the pragmatic gathering of these data, yet these methods rely on the ability of staff to identify and verbalise the ways in which they succeed. To increase the likelihood of achieving theoretical saturation, where appropriate, the researchers will conduct several focus groups in each setting (eg, on multiple wards in large hospital specialties and in dual-sited general practices). They will also discuss the emerging data and concurrently collect and analyse the data in order to minimise the effects. Finally, although this study adopts aspects of the positive deviance framework,[24] it does not apply

the approach in its entirety. The third and fourth stages of the framework (testing positively deviant hypotheses and disseminating them to others) will not be conducted during the PACT programme of work as we seek to develop an intervention on the basis of patient experiences of involvement and the existing literature, not only staff perceptions of safety.

It is envisaged that this study will help to identify practices and policies that may improve the safety and quality of transitions of care. By sampling healthcare professionals from across the care continuum, including secondary, community and general practice teams, we hope to better understand the disconnect that exists between services and how staff adjust to minimise their effects. Some of the challenges and solutions uncovered may not be amenable to change at a local level or through the PACT intervention and so broader system and policy level changes may be required.

**Acknowledgements** This research is supported by the NIHR Collaboration for Leadership in Applied Health Research and Care, Yorkshire and Humber—Evidence-Based Transformation Theme.

**Contributors** RL, JoH, RF, LS, AC and JW were involved in the design of the overall PACT programme of research and the conception of studies within it. RB developed the protocol for the current study and JoH, JM, LS, AC, RF, JW and RL contributed to the study design. RB wrote a first draft of the manuscript and all authors have contributed to the drafting, reviewing of it. All authors have approved the final version of this manuscript for submission.

**Funding** This report is independent research funded by the National Institute for Health Research (National Institute for Health Research Programme Grants for Applied Health Research, Partners at Care Transitions (PACT): improving patient experience and safety at transitions in care, RP-PG-1214- 20017).

**Disclaimer** The views expressed in this publication are those of the authors and not necessarily those of the NHS, the National Institute for Health Research or the Department of Health.

**Competing interests** None declared.

**Patient consent** Not required.

**Ethics approval** The preparatory work did not require ethical or HRA approval but permissions to access the readmission data were granted by the relevant CCGs and the North East Commissioning Support Unit. Ethics and HRA approvals for the qualitative data collection were gained separately for primary and secondary care: (1) University of Leeds ethical approval: primary care reference—17-0202 , date 18 July 2017; secondary care reference—17-0234, date 30 August 2017; (2) HRA approval: primary care IRAS reference—230156, date 24 August 2017; secondary care IRAS reference 233797, date 22 September 17. Local NHS capability and capacity approvals were granted by all organisations involved; (3) The study was registered on the UK Clinical Research Network Study Portfolio: primary care reference—35272; secondary care reference—36174.

**Provenance and peer review** Not commissioned; externally peer reviewed.

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
