## [Reviewer comments · BMJ Open]

ARTICLE DETAILS

TITLE (PROVISIONAL)	Partners at Care Transitions - exploring healthcare professionals' perspectives of excellence at care transitions for older people: a study protocol
AUTHORS	Baxter, Ruth; O'Hara, Jane; Murray, Jenni; Sheard, Laura; Cracknell, Alison; Foy, Robbie; Wright, John; Lawton, Rebecca

VERSION 1 – REVIEW

REVIEWER	Nancy Schoenborn Johns Hopkins University USA
REVIEW RETURNED	30-Mar-2018

GENERAL COMMENTS	Overall comments: The authors describe the study protocol for a qualitative study to explore how staff within high performing teams, both in general practices and hospital specialties, are able to achieve their high performance regarding care transitions. High performance is defined as low or improving readmission rates. The project is innovative and will yield valuable results to advance the field. The overall description of the protocol is clear. 1. Background: the discussion of "ongoing care needs" of patients upon leaving the hospital (line 11) can probably be clarified and expanded upon. Arguably older patients with chronic illnesses will always have ongoing care needs, I believe the authors are referring to a higher than baseline level of care that's directly related to the recent hospitalization, and may consider rewording as such.2. Methods: I find the description of the Preparatory work to identify high performing teams to be helpful. I'd like to understand better why the control limits were set at 2 versus 3 sigma for general practice versus hospital specialty, as opposed to say choosing 2 sigma or 3 sigma for both groups? (page 6, lines 44-46).3. I'd also like for the authors to describe in more detail how the authors defined improved performance over time (page 6, lines 46-49) . i.e. how much improvement over how much time would qualify a practice or specialty to be in this category?4. Methods: Since the study goal is to identify not only explicit content but also abstract content such as team dynamics, culture etc, the results may be subject to an extra layer of subjectivity and variation depending on the specific researcher who is doing the observation and the interview. The authors should provide more details on the qualifications and number of research team members who will be doing the actual interviews, focus groups, and observation of staff meetings, and assuming there is more than one person who will be doing these tasks, how to ensure consistency in their approaches.5. Discussion – the study scope is ambitious, and involves a large number of study participants for a qualitative study. The stratification by practice type (general practice versus hospital specialty) is very much appropriate, and the described sampling strategy is reasonable. However, that still involves interviewing a very wide range of multidisciplinary staff at any single site, and then the sites can vary in multiple ways depending on geography, patient mix, and local
--

	support/health systems. I worry about truly reaching “thematic saturation” and also not sure whether a “typical” participant will emerge from this highly heterogeneous mix. As it is not possible to change the study methods, this point should be discussed in the last part of discussion section – i.e. what are strategies to ameliorate this.
--	--

REVIEWER	Christina R. Whitehouse, PhD, CRNP, CDE University of Pennsylvania School of Nursing NewCourtland Center for Transitions and Health
REVIEW RETURNED	24-May-2018

GENERAL COMMENTS	This is an extremely well written protocol. There is a vast amount of literature focusing on transitions in care however there has been little attention to exploring practices successful in care transitions. Using a positive deviance approach can provide information crucial to successful transitions within a healthcare organization. I look forward to seeing the results in the literature. It was a pleasure to review this manuscript.
---

VERSION 1 – AUTHOR RESPONSE

Response to reviewers: Manuscript ID bmjopen-2018-022468

Reviewers Comments	Response	Page
Editor:		
Please provide more details related to the timeline and current status of the study.	We have provided further information regarding the timelines for the study in the 'Methods - Study design and status' section. The preparatory work described above took place between June and August 2017. The study commenced in September 2017 and site recruitment is due to be completed by the end of May 2018. Data collection and analysis will be iterative and flexible according to the needs of the study.	8
Reviewer 1:		
Background: the discussion of “ongoing care needs” of patients upon leaving the hospital (line 11) can probably be clarified and expanded upon. Arguably older patients with chronic	Many thanks for your positive appraisal of this manuscript and constructive comments to support its improvement. Indeed, we did intend for ‘ongoing care needs’ to refer to those that result from a	4

illnesses will always have ongoing care needs, I believe the authors are referring to a higher than baseline level of care that's directly related to the recent hospitalization, and may consider rewording as such.	hospital admission and a shorter length of stay. We have rephrased the sentence to provide clarification and have given additional detail for each the examples. The sentence now reads: However, shorter lengths of stay mean patients often leave hospital with ongoing care needs that are directly related to their recent hospitalisation, such as wound care treatment, changes to, or monitoring of, new medications, and limited mobility associated with deconditioning and inactivity.	
Methods: I find the description of the Preparatory work to identify high performing teams to be helpful. I'd like to understand better why the control limits were set at 2 versus 3 sigma for general practice versus hospital specialty, as opposed to say choosing 2 sigma or 3 sigma for both groups? (page 6, lines 44-46).	Control limits were set at 2 and 3 sigma for the analyses of the both the general practice and hospital specialty data. We have amended the sentence to provide clarity: In both settings, control limits were set at 2 and 3 sigma (σ) and high performers were classed as those teams (general practices or hospital specialties) that exceeded these limits.	6
I'd also like for the authors to describe in more detail how the authors defined improved performance over time (page 6, lines 46-49) . i.e. how much improvement over how much time would qualify a practice or specialty to be in this category?	We have added in the following sentence to address this comment: In addition to learning from those who performed the best in the region, bar charts were also plotted using the secondary care data to identify hospital specialties with reducing readmission rates, i.e. those with improved performance over time. It was difficult to apply definitive rules to this as data varied markedly between hospital sites and specialties. As such, the researchers discussed the data as a group and selected the high performing hospital specialties that	6

	demonstrated the greatest reduction in readmission rates over the three year period.	
Methods: Since the study goal is to identify not only explicit content but also abstract content such as team dynamics, culture etc, the results may be subject to an extra layer of subjectivity and variation depending on the specific researcher who is doing the observation and the interview. The authors should provide more details on the qualifications and number of research team members who will be doing the actual interviews, focus groups, and observation of staff meetings, and assuming there is more than one person who will be doing these tasks, how to ensure consistency in their approaches.	The additional information below has been added into the 'Methods – Procedure' section: Where possible, all focus groups will be facilitated by RB who has a background in health services researcher and is educated to doctorate level. In each setting RB will be supported Where possible, all focus groups will be facilitated by RB who has a background in health services researcher and is educated to doctorate level. In each setting RB will be supported by a researcher who has a clinically relevant background - an occupational therapist, a general practitioner, and a community nurse. This will facilitate the collection of data from insider and outsider perspectives. All researchers will help to develop the focus group schedules to ensure consistency in their data collection approaches. Researchers will meet regularly to discuss the emerging data.	9
Discussion – the study scope is ambitious, and involves a large number of study participants for a qualitative study. The stratification by practice type (general practice versus hospital specialty) is very much appropriate, and the described sampling strategy is reasonable. However, that still involves interviewing a very wide range of multidisciplinary staff at any single site, and then the sites can vary in multiple ways depending on geography, patient mix, and local support/health	We too recognise that the scope of this study is ambitious. However, we seek to try and balance the need to achieve theoretical saturation with our aim of gathering data across heterogeneous clinical settings. We believe that exploring the similarities and differences between different settings will add to the current literature while also supporting our future work to develop the PACT intervention (which will be implemented across a variety of different clinical settings). We have added a paragraph into the 'Discussion – Strengths and limitations'	13

systems. I worry about truly reaching “thematic saturation” and also not sure whether a “typical” participant will emerge from this highly heterogeneous mix. As it is not possible to change the study methods, this point should be discussed in the last part of discussion section – i.e. what are strategies to ameliorate this.	section to address this comment: A key strength of this study is the inclusion of multiple perspectives across a variety of healthcare settings to explore how different clinical teams successfully support transitions of care. However, this heterogeneous sample may limit our ability to achieve theoretical saturation. Similarly, focus group and interview methods were chosen to facilitate the pragmatic gathering of these data, yet these methods rely on the ability of staff to identify and verbalise the ways in which they succeed. To increase the likelihood of achieving theoretical saturation, where appropriate, the researchers will conduct several focus groups in each setting (e.g. on multiple wards within in large hospital specialties and in dual sited general practices). They will also discuss the emerging data and concurrently collect and analyse the data in order to minimise the effects.	
Reviewer 2:		
This is an extremely well written protocol. There is a vast amount of literature focusing on transitions in care however there has been little attention to exploring practices successful in care transitions. Using a positive deviance approach can provide information crucial to successful transitions within a healthcare organization. I look forward to seeing the results in the literature. It was a pleasure to review this manuscript.	We thank the reviewer for their kind comments and look forward to being able to share the results with them.